# Timeline of Developmental Defects Generated upon Genetic Inhibition of the Retinoic Acid Receptor Signaling Pathway

**DOI:** 10.3390/biomedicines11010198

**Published:** 2023-01-12

**Authors:** Marius Teletin, Manuel Mark, Olivia Wendling, Nadège Vernet, Betty Féret, Muriel Klopfenstein, Yann Herault, Norbert B. Ghyselinck

**Affiliations:** 1Institut de Génétique et de Biologie Moléculaire et Cellulaire (IGBMC), Centre National de la Recherche Scientifique (CNRS UMR7104), Institut National de la Sante et de la Recherche Médicale (INSERM U1258), Université de Strasbourg (UNISTRA), 1 Rue Laurent Fries, BP-10142, F-67404 Illkirch Graffenstaden, France; 2Service de Biologie de la Reproduction, Hôpitaux Universitaires de Strasbourg (HUS), F-67000 Strasbourg, France; 3Institut Clinique de la Souris (ICS), Université de Strasbourg, CNRS, INSERM, CELPHEDIA, PHENOMIN, 1 Rue Laurent Fries, 67404 Illkirch Graffenstaden, France

**Keywords:** mouse, heart development, lung development, eye development, inner ear development, axial rotation, embryonic turning, cardiac looping, HREM

## Abstract

It has been established for almost 30 years that the retinoic acid receptor (RAR) signalling pathway plays essential roles in the morphogenesis of a large variety of organs and systems. Here, we used a temporally controlled genetic ablation procedure to precisely determine the time windows requiring RAR functions. Our results indicate that from E8.5 to E9.5, RAR functions are critical for the axial rotation of the embryo, the appearance of the sinus venosus, the modelling of blood vessels, and the formation of forelimb buds, lung buds, dorsal pancreatic bud, lens, and otocyst. They also reveal that E9.5 to E10.5 spans a critical developmental period during which the RARs are required for trachea formation, lung branching morphogenesis, patterning of great arteries derived from aortic arches, closure of the optic fissure, and growth of inner ear structures and of facial processes. Comparing the phenotypes of mutants lacking the 3 RARs with that of mutants deprived of all-*trans* retinoic acid (ATRA) synthesising enzymes establishes that cardiac looping is the earliest known morphogenetic event requiring a functional ATRA-activated RAR signalling pathway.

## 1. Introduction

All-trans retinoic acid (ATRA), the active metabolite of vitamin A, is synthesized in its target tissues through the activity of retinaldehyde dehydrogenases (ALDH1A1, ALDH1A2, ALDH1A3) and acts through binding to retinoic acid receptors (RARA, RARB and RARG; encoded by the *Rara*, *Rarb,* and *Rarg* genes), which are ligand-inducible transcription regulators [1].

That signalling through RAR regulates gene networks involved in growth, morphogenesis, and cellular differentiation was inferred from the phenotypic analysis of mice carrying null alleles of 2 RAR isotypes from the one-cell stage embryo onwards [knock-out (KO) mutants]. These *Rara*^−/−^;*Rarb*^−/−^, *Rara*^−/−^;*Rarg*^−/−^ and *Rarb*^−/−^;*Rarg*^−/−^ KO mutants display a large array of congenital malformations, affecting the cardiovascular, respiratory, and urogenital systems, as well as the brain, sense organs, ventral body wall, forelimbs, and craniofacial structures [2]. 

To clarify when the RAR signalling pathway is required for development, we designed a temporally controlled genetic ablation procedure based on the use of a ubiquitously expressed recombinase (cre/ERT^2^) that can be activated by tamoxifen (TAM) to conditionally invalidate the *Rara* and *Rarg* genes in the *Rarb*^−/−^ background [3]. This approach previously allowed us to demonstrate that the RAR signalling pathway is required for many developmental processes that are determined between embryonic day 10.5 (E10.5) and E11.5 [4]. Here, by analysing the phenotypic consequences of ablation of all 3 *Rar* genes at E8.5 and E9.5 on 3-dimensional reconstructions from high-resolution episcopic microscopy (HREM) images, we define critical windows of time during which the RAR signalling is required during early development for proper organogenesis.

## 2. Materials and Methods

### 2.1. Mice 

Mice were on a mixed C57BL/6 (50%)/129/SvPass (50%) genetic background. The procedure for conditional ablation of *Rar*-coding genes has been described [3]. In brief, females homozygous for L2 alleles of *Rara* and *Rarg* and KO for *Rarb* (i.e., *Rara*^L2/L2^;*Rarg*^L2/L2^;*Rarb*^−/−^) were mated with males bearing one copy of the ubiquitously expressed *Tg*(*Ubc-cre/ERT2*) transgene [5] and homozygous for L2 alleles of *Rara* and *Rarg* and KO for *Rarb* [i.e., *Tg*(*Ubc-cre/ERT2*);*Rara*^L2/L2^;*Rarg*^L2/L2^;*Rarb*^−/−^]. Noon of the day of a vaginal plug was taken as embryonic day 0.5 (E0.5). To activate the cre/ERT^2^ recombinase in embryos, a single TAM treatment was administered to the pregnant females by oral gavage at E7.5, at E8.5, or at E9.5 (120 mg/kg body weight). This resulted in embryos KO for *Rarb*, in which *Rara* and *Rarg* were ablated upon TAM administration when they were bearing *Tg*(*Ubc-cre/ERT2*) (hereafter referred to as mutants), as well as their control littermates when the embryos or foetuses were free of *Tg*(*Ubc-cre/ERT2*) (hereafter referred to as controls). 

### 2.2. Collection, Staging, Tissue Processing and Reconstruction of Embryos

Since the complete loss of *Rara* and *Rarg* was assessed as efficient 24 h after TAM administration by immunochemistry and western blotting [3], embryos treated with TAM at E7.5, E8.5, and E9.5 are referred to as *Rarabg^ΔE8.5^*, *Rarabg^ΔE9.5^* and *Rarabg^ΔE10.5^* mutants, respectively. *Rarabg^ΔE8.5^* mutants were collected for analyses at E9.5 and E10.5. *Rarabg^ΔE9.5^* mutants were collected at intervals of 24 h from E10.5 to E12.5. *Rarabg^ΔE10.5^* mutants were collected at E11.5, E12.5, and E14.5 (Figure 1). Embryos were fixed for 24 h in Bouin’s fluid. At inspection, we selected the pairs of mutant and control littermates that were closest in terms of general size and external features [6] (e.g., in Figure 2a–f). For HREM analysis, embryos were dehydrated and embedded in methacrylate resin (JB-4, Polysciences) containing eosin and acridine orange. After polymerisation and hardening, the resin blocks were used for HREM data generation [7,8]. Section thickness was set at 5 μm (E9.5, E10.5, E11.5, and E12.5) and 7 μm (E14.5). HREM images were loaded into Fiji [9] to generate virtual stacks. 2D images were segmented manually with 3D Slicer [10], without interpolation between sections. 

## 3. Results

### 3.1. Abnormal Phenotype of Rarabg^ΔE9.5^ Embryos and Comparison with Rarabg^ΔE10.5^ Embryos 

Almost all *Rarabg^ΔE9.5^* mutants, lacking the 3 *Rar* genes from E9.5 onwards, died between E12.5 and E13.5. They were analysed at E10.5, E11.5, and E12.5, and compared with control littermates and to *Rarabg^ΔE10.5^* mutants (Table 1; Figure 2). As anticipated, *Rarabg^ΔE9.5^* mutants reproduced all the defects present in *Rarabg^ΔE10.5^* mutants at the same developmental stages [4]. They also displayed additional abnormalities which are detailed below.

#### 3.1.1. Tracheal Agenesis 

From E10.0 to E11.5, the anterior foregut is divided by a longitudinal septum into two tubes, the esophagus on the dorsal side and the trachea on the ventral side [16,17,18]. 

In control embryos at E10.5, the first sign of division of the foregut appeared as a protuberance on the ventral wall of the foregut at the site of bifurcation of the stem bronchi (i.e., the presumptive carina; Figure 3a). The division process was completed at E11.5 (Figure 3b; Appendix A). In *Rarabg^ΔE9.5^* mutants at E10.5, E11.5, and E12.5, no separation occurred. Instead, a single foregut tube was present with the lung buds arising directly from the walls of this tube (Figure 3d–f; Appendix A).

In *Rarabg^ΔE10.5^* mutants, the trachea was normally formed at E11.5 (Figure 3g). Moreover, it was completely separated from the esophagus in all *Rarabg^ΔE10.5^* mutants previously analysed at E14.5 on histological sections [4]. These observations indicate that the RAR signalling pathway is needed for the separation of the foregut at developmental stages between E9.5 and E10.5, i.e., around the onset of this developmental process.

#### 3.1.2. Delayed Lung Branching Morphogenesis

Lungs emerge at E9.5 as two endodermal buds from the ventrolateral walls of the foregut, and subsequently, from E10.5 to E16.5, they undergo repetitive branching to generate the bronchial (or airway) tree [16,19]. The pattern of lung branching occurs in a precise spatio-temporal sequence and is highly reproducible [20].

Control embryos at E10.5 displayed unbranched right and left lung buds (Figure 3a). At E11.5, the right stem bronchus had generated a series of four secondary (lobar) bronchi, whereas the left stem bronchus displayed a single secondary branch (Figure 3b; Appendix A). At E12.5, both the left and right lungs had acquired tertiary (segmental) bronchi (Figure 3c; Appendix A). 

Lung budding proceeded normally in the vast majority (8 out of 9) of the *Rarabg^ΔE9.5^* mutants (Figure 3d–f), with one exception at E10.5 in which the left lung bud was missing (not shown). Then, lung branching was delayed in all *Rarabg^ΔE9.5^* mutants analysed at E11.5 and E12.5. At E11.5, the right stem bronchus had either no secondary branch (2 out of 3) or a single small secondary branch, and the left stem bronchus was never branched (Figure 3e; Appendix A). At E12.5, the right lung of the mutants displayed a reduced number of branches; the left primary bronchus was short and did not display any secondary branch (Figure 3f; Appendix A). 

*Rarabg^ΔE10.5^* mutants at E11.5 (Figure 3g) and E12.5 (Figure 3h) exhibited a milder delay in lung branching than *Rarabg^ΔE9.5^* mutants at the same developmental stages. Interestingly, lung branching may be, at least partially, recovered as development proceeds, as its delay was detected in only 1 (out of 5) mutant analysed on histological sections at E14.5 [4]. 

In summary, when the 3 *Rars* are invalidated at E9.5, the development of the primary lung buds occurred normally up to E10.5, but subsequently, the appearance of secondary branches was delayed for 24 h on the right side and for at least 48 h on the left side. Although less marked, the delay was also observed when the *Rars* are invalidated around E10.5. These data indicate that signalling through RAR normally promotes the divisions of the lung buds that give rise to the secondary bronchi.

#### 3.1.3. Persistent Truncus Arteriosus

In early embryos, the blood is conveyed from the heart into the arterial system via the aortic sac, a non-myocardial cavity embedded in the pharyngeal mesenchyme. Between E11.5 and E12.5, the aorticopulmonary septum, a protrusion from the wall of the aortic sac between the 4th and 6th aortic arches, divides the sac in 2 separate vessels, the ascending aorta and the pulmonary trunk [21,22]. Both vessels were always identified at E12.5 in control embryos (Figure 3c; Appendix A). In contrast, the 3 *Rarabg^ΔE9.5^* mutants at E12.5 showed no evidence of a pulmonary trunk and displayed single cardiac outflow vessel, or persistent truncus arteriosus (PTA; Figure 3f; Appendix A).

In the *Rarabg^ΔE10.5^* mutants analysed at E12.5, the septation of the aortic sac was complete (Figure 3h), and only 1 out of the 5 mutants previously analysed at E14.5 displayed a PTA [4]. Taken together, these data indicate that the PTA is determined between E9.5 and E10.5, i.e., at least 24 h prior to the morphological onset of aortic sac septation. 

#### 3.1.4. Ageneses of the 3rd, 4th, and 6th Aortic Arches 

Aortic arches (numbered 1, 2, 3, 4, and 6) are paired arteries which develop in a cranial to caudal sequence between E8.5 and E10.0 to transmit the blood from the aortic sac to the left and right dorsal aortas. In normal mouse embryos at E10.5, the 1st and 2nd aortic arches have degenerated, whereas the 3rd, 4th, and 6th aortic arches are bilaterally symmetrical and they remain so at E11.5. They do not substantially remodel until the E12.5 when the right 6th aortic arch involutes. Subsequently, between E12.5 and E14.5, the set including the aortic arches and the dorsal aortas becomes asymmetrical due to the regression of specific arterial segments. Through this vascular remodeling process, major cephalic arteries are formed: left and right common carotid arteries (from the 3rd aortic arches); proximal segment of the right subclavian artery (from the right 4th aortic arch); definitive aortic arch (from the left 4th aortic arch and left dorsal aorta); and arterial duct (from the left 6th aortic arch) [23,24]. 

In control embryos at E10.5 and E11.5, both left and right dorsal aortas received blood via the ipsilateral 3rd, 4th, and 6th arches, and the 1st and 2nd aortic arches had regressed, as expected (Table 2; Figure 3a,b; Figure 4d; Appendix A). In contrast, all *Rarabg^ΔE9.5^* mutants at E10.5 and E11.5 (n = 6) had no 6th arches (Table 2; Figure 3d,e; Figure 4e,f; Appendix A). The 3rd and/or 4th aortic arches were often absent or incomplete (Table 2; Figure 3e; Figure 4f) and the right or left dorsal aorta was occasionally hypoplastic (Figure 4f). Additionally, two of the mutants at E10.5 showed bilaterally persistent 1st and/or 2nd aortic arches, probably reflecting an attempt to functionally compensate the deficiency in the more caudal arches (Table 2; Figure 4f). Of the 3 *Rarabg^ΔE9.5^* mutants analysed at E12.5, two were developing a definitive arch of the aorta on the right side while the third mutant had a right retroesophageal subclavian artery (Figure 3f; Appendix A). 

*Rarabg^ΔE10.5^* mutants at E11.5 and E12.5 showed normal, aortic arch patterns (Figure 3g,h). Moreover, in a majority of the *Rarabg^ΔE10.5^* mutants at E14.5 (4 out of 5), the pattern of the cephalic arteries was normal [4]. The only exception was the mutant with a PTA which lacked the arterial duct. Taken together, these data indicate that the RAR signalling pathway is instrumental to the formation of the 3rd, 4th, and 6th aortic arches between E9.5 and E10.5.

#### 3.1.5. Reduced Development of the 3rd and 4th Pharyngeal Pouches 

Pharyngeal pouches (numbered 1, 2, 3, and 4) are bilateral outgrowths of the pharyngeal endoderm that arise sequentially, from anterior to posterior. The most anterior pouch (PP1) forms at E8.5, followed by the 2nd pouch (PP2) at E9.0, the 3rd pouch (PP3) at E9.5, and lastly the 4th pouch (PP4) at E10.0. Subsequently, from E11.5 onwards, the 3rd pouch participates in the development of the thymus and parathyroid glands and the 4th pouch gives rise to the calcitonin-producing cells [25,26].

In control embryos, the complete series of pharyngeal pouches (PP1 to PP4) was present at E10.5 and E11.5 (Table 2; Figure 3a,b; Figure 4a,d; Appendix A). In contrast, *Rarabg^ΔE9.5^* embryos always lacked the 4th pouch at E10.5 and E11.5 (Table 2; Figure 3d,e; Figure 4b,c,e,f; Appendix A). Additionally, the *Rarabg^ΔE9.5^* embryos with absent or incomplete 3rd aortic arches also displayed an absence or hypoplasia of the 3rd pharyngeal pouch on the same side as the arterial defect (Table 2; Figure 4c,f). 

*Rarabg^ΔE10.5^* mutants analysed at E11.5 and E12.5 displayed normal pouches (Figure 3g). Moreover, the size and position of the thymus were normal in all *Rarabg^ΔE10.5^* mutants analysed at E14.5 on serial histological sections [4]. Taken together, these data indicate that signalling through RARs is required for the formation of 3rd and 4th pouches at developmental stages between E9.5 and E10.5. 

#### 3.1.6. Absence of Closure of the Optic Fissure and Associated Ocular Defects 

Development of the retina begins at E9.5 with the formation of an optic vesicle which, from E9.5 to E10.5, folds inwards to form a double-layered optic cup with a fissure (the optic fissure) on its ventral surface. Between E10.5 and E12.5, the margins of the optic fissure grow towards each other until they fuse, leaving a small opening for the blood vessels and optic nerve, the optic disc. A defect in this process of closure of the optic fissure leaves a gap (or coloboma) which can be extended throughout the optic cup or be restricted either to the iris or the optic disc [27]. 

In control embryos, the closure of the optic fissure was complete at E12.5 (Figure 5a–c; Appendix A). In contrast, in *Rarabg^ΔE9.5^* mutants at E12.5 the ventral margins of the optic cup remained separated, resulting in a complete, bilateral, coloboma of the retina (white arrowhead, Figure 5d–f; Appendix A). Other ocular defects, involving the retina and its surrounding mesenchyme, were consistently associated with the retinal coloboma, including: (i) shortening of the ventral portion of the retina (Figure 5e,f), (ii) malposition of the optic cup and the lens, which were tilted ventrally (compare Figure 5b,e), and (iii) cryptophthalmos (Figure 2f). In this latter abnormality, the cornea and eyelid folds were not identifiable and were replaced by a thick layer of mesenchyme interposed between the surface ectoderm and the optic cup (not shown). 

*Rarabg^ΔE10.5^* mutants at E12.5 (Figure 5g–i) and E14.5 [4] displayed a coloboma which was restricted to the optic disk. They also showed a malposition of the optic cup and the lens, a shortening of the ventral retina, and a cryptophthalmos which were as severe as in *Rarabg^ΔE9.5^* mutants (Figure 4). These data indicate essential roles of RAR signalling in ocular development between E9.5 and E11.5.

#### 3.1.7. Hypoplasia of the Nasal Processes and Cavities 

The face develops from the coordinated growth of five primordia: the frontonasal process and the paired maxillary and mandibular processes of the first pharyngeal arches. By E10.5, the frontonasal process gives rise to paired medial nasal and lateral nasal processes flanking each of the nasal pits (Figure 6a,c). Then, from E11.5 to E12.5, the medial nasal processes gradually merge together to form the intermaxillary segment of the embryonic face [28] (Figure 6c,e).

In *Rarabg^ΔE9.5^* mutants at E10.5, the lateral and medial nasal processes were markedly hypoplastic and the nasal pits faced laterally, whereas in the control embryos, they were oriented medially (compare Figure 6a,b). At E11.5, the edges of medial nasal processes were separated by a wide groove, whereas in control embryos, these processes had started to merge together at the midline (compare Figure 6c,d). At E12.5, the intermaxillary segment of the mutants was truncated and displayed a median cleft as well as widely spaced nasal pits (compare Figure 6e,f). 

*Rarabg^ΔE10.5^* mutants at E11.5 and E12.5 were undistinguishable from age-matched control littermates with respect to the facial processes (not shown). At E14.5, they only displayed a mild shortening of snout (which derives from the intermaxillary segment), without any facial cleft nor excessive spacing of the nostrils (compare Figure 7a–d). Altogether, these observations indicate that the RAR signalling pathway is needed for the correct formation of the external aspect of the midface at developmental stages between E9.5 and E10.5.

The nasal cavities arise from the invagination of the nasal pits into the midfacial mesenchyme. They expand and progress towards the oral cavity from which they remain separated by the oronasal membrane. This membrane disintegrates at E12.5, leading to communications between the nasal cavities posterior to the primary palate and the oral cavity, also named the primitive choanae [29] (Figure 5a; Appendix A). In *Rarabg^ΔE9.5^* mutants, the impairment of nasal cavities development and the lack of communication with the oropharyngeal cavity led to hypoplasia of the nasal cavities and to choanal atresia, respectively (Table 1; Figure 5d; Appendix A). Very similar defects were found in *Rarabg^ΔE10.5^* mutants (Figure 5g). In contrast, these defects were absent in *Rarabg^ΔE11.5^* mutants [3]. This chronology implies that E10.5 to E11.5 spans the critical developmental period during which RAR signalling is required for the development of the nasal cavities. 

#### 3.1.8. Near Normal Morphogenesis of the Inner Ear

The mouse inner ear originates at E8.0 in the form of the otic placode, formed from a thickening of the head ectoderm, lateral to the hindbrain. This otic placode then invaginates into the mesenchyme to form the otocyst (otic vesicle) at E9.5 (Figure 8a, inset). At E10.5, the otocyst develops a short endolymphatic duct (Figure 8c, inset); then, during the next 72 h, it is converted though a series of evaginations and tissue appositions into a labyrinth of inter-connected ducts and chambers (i.e., the membranous labyrinth) [30,31].

In *Rarabg^ΔE9.5^* and *Rarabg^ΔE10.5^* mutants at E12.5, the main subdivisions of the definitive inner ear, including the endolymphatic duct, utricle, saccule, and semicircular canals, were established and undistinguishable from those of control embryos (compare Figure 8a,b; and not shown). We also analysed *Rarabg^Δ10.5^* mutants at E14.5, i.e., at the near end of the morphogenetic process generating the mature inner ear. The endolymphatic sac and the saccule appeared smaller in the mutants, but apart from these localized growth deficiencies, we did not observe major differences with the control littermates (compare Figure 8c,d; Appendix A). Taken together, these results indicate that the RAR signalling pathway is essentially dispensable for inner ear morphogenesis from the otocyst stage onward. 

### 3.2. The Overall Development of Rarabg^ΔE8.5^ Mutants Is Arrested Shortly after Rar Excision

*Rarabg^ΔE8.5^* mutants, lacking the 3 *Rar* genes from E8.5 onwards, died at E11.5 at the latest. They were analysed at E9.5 and E10.5.

*Rarabg^ΔE8.5^* mutants collected at E9.5 (n = 3) and E10.5 (n = 9) were alive, as judged from the presence of heart beats, and displayed similar external features. None of them had undergone the axial rotation (embryonic turning) that normally occurs between E8.5 and E9.0 [6,32] (compare Figure 9a,b; Appendix A). Their anterior region was comparable to control embryos at E9–early E9.5 in terms of closure of the forebrain and development of the 1st pharyngeal arch. In contrast, their posterior region was severely shortened, remained open ventrally, and showed abnormally small and densely packed somites. Other external defects included: absence of the limb buds and of the 2nd pharyngeal arches and dilation of the pericardial cavity (Table 3; Figure 9b). Two out of twelve *Rarabg^ΔE8.5^* mutants displayed an open neural tube at the hindbrain level (not shown). 

HREM analysis of the 3 *Rarabg^ΔE8.5^* mutants at E9.5 revealed that the neural tube was narrow and had extensively folded walls (Figure 9b). Multiple cell debris were seen in the lumen of the forebrain. The 2nd pharyngeal pouch was always absent and the 2nd aortic arch was missing on one or both sides (Figure 9b). The mutant heart showed considerably dilated cavities in the outflow tract, the primitive ventricle, and the primitive atrium (Figure 9b and Figure 10c,d). The latter, as opposed to the control atrium (Figure 10a,b), did not receive blood inflow from the sinus venosus, but instead showed an abnormal communication (aorto-atrial fistula) with either the left or the right dorsal aortas [35] (Figure 10d). 

The sinus venosus is a paired structure produced, around E9.0, by the confluence of the main venous channels (the cardinal, umbilical, and vitelline veins). It comprises right and left ducts (called “horns”), each collecting the blood from one side of the embryo and transferring it to the primitive atrium [36] (Figure 9a and Figure 10a,b). All 3 *Rarabg^ΔE8.5^* mutants lacked the sinus venosus and thus displayed no connection between the heart and the cardinal and umbilical veins; the vitelline veins were not identifiable (compare Figure 9a,b and Figure 10a–d; Appendix A). In contrast, the sinus venosus was normal in *Rarabg^ΔE9.5^* mutants analysed at E10.5 (not shown).

Analysis of *Rarabg^ΔE8.5^* at E10.5 was unachievable due to a general collapse of all the blood vessels and to multiple necrotic foci in many tissues. Altogether, these observations indicate that these mutants display a completely penetrant lethal phenotype, characterised by an absence of axial rotation, an arrest in morphogenesis and body growth around E9.0, and a set of severe malformations of the neural tube, heart, and blood vessels. This phenotype is strikingly similar to that displayed by *Aldh1a2^−/−^* KO mutants, which are devoid of ATRA-signalling activity [33,34,37] (Table 3).

### 3.3. Incomplete Excision of Rara and Rarg at E8.5 Yields Severe Developmental Defects

Within litters exposed to TAM at E7.5 and analysed at E10.5, we found embryos that had undergone axial rotation and were externally similar to E10.5 control embryos except for the forelimb buds which were small and malformed. These embryos happened to be incompletely excised (iΔ) for both *Rara* and *Rarg,* and were referred to as *Rarabg^iΔE8.5^* mutants. 

As anticipated, *Rarabg^iΔE8.^*^5^ mutants (n = 3) had defects in common with *Rarabg^ΔE9.5^* mutants as, for example, the hypoplasia of the nasal processes, the malposition of the nasal pit, the abnormal aortic arch patterns, and the agenesis of the 4th pharyngeal pouch (Figure 11a,b). They also all displayed additional abnormalities attesting to the earlier excision of the *Rars*, including: forelimb bud hypoplasia (Figure 11a); bilateral agenesis of the lung buds and agenesis of the dorsal pancreatic bud (Figure 11b); small otocysts, supernumerary otocysts, and uni- or bilateral absence of the lens (compare Figure 11c,d). 

## 4. Discussion

### 4.1. RARs Have Essential Roles of in the Establishment of the Respiratory System

We provide here the genetic evidence that, in the respiratory system, the RAR signalling pathway sequentially controls major morphogenetic events at different time points: lung budding from E8.5 to E9.5, formation of the trachea from E9.5 to E10.5, and lung branching morphogenesis from E9.5 to E11.5 (Figure 12a).

The phenotype of *Rarabg^iΔE8.^*^5^ mutants indicates that RARs are also instrumental to the outgrowth of the left and right lung buds from the foregut. This agrees with previous observations showing that this budding process can be inhibited by pharmacologically blocking the RAR signalling pathway [38]. Taken together, these data strongly support the possibility that agenesis of the left lung, a near constant feature of *Rara*^−/−^;*Rarb*^−/−^ KO mutants, results from the absence of left lung budding ((Table 1). They also suggest that the constant presence a right lung in the *Rara*^−/−^;*Rarb*^−/−^ KO mutants, as opposed to the absence of the right lung bud in *Rarabg^iΔE8.^*^5^ mutants, reflects a functional compensation by RARG in the KO mutants.

All *Rara*^−/−^;*Rarb*^−/−^ KO mutants display bilateral lung hypoplasia [11,12], a defect which is typically caused by failure of lung branching [19]. We show here that RAR promotes the first step in lung branching, namely the formation of the secondary branches from the primary lung buds. In addition, (i) *Rarabg^ΔE11.5^* mutants have normal lungs [4] and (ii) the development of the bronchial tree cannot be inhibited by pharmacologically blocking the RARs after E11.75 [38]; the critical time-window during which RARs promote lung branching occurs between E9.5 and E11.5 (Figure 12a). Therefore, the RAR signalling pathway stimulates the formation of bronchi only at an early stage of lung morphogenesis, which corresponds to a narrow period only, given that the bronchial tree develops until E16.5 [16,19].

### 4.2. RARs Exert Functions at Different Stages of Cardiovascular Development

The phenotypic analysis of *Rarabg^ΔE8.^*^5^, *Rarabg^ΔE9.^*^5^ and *Rarabg^ΔE10.^*^5^ mutants indicate that the RAR signalling pathway plays essential roles between E8.5 and 9.5 in the formation of the sinus venosus; then, between 9.5 and 10.5, in the partitioning of the aortic sac and formation of the 3rd, 4th, and 6th aortic arches (Figure 12b).

The dramatic vascular defects observed in *Rarabg^ΔE8.5^*, including the absence of the sinus venosus, the vitelline veins, and the aorto-atrial fistulae, is in keeping with the fundamental role of ATRA in blood vessel remodeling at this developmental stage [34,39]. As already mentioned, all abnormalities of *Rarabg^ΔE8.5^* mutants, including the absence of the sinus venosus, are strikingly similar to those displayed by *Aldh1a2^−/−^* KO mutants (Table 3). The only notable difference lies in the relative position of the heart chambers. In *Rarabg^ΔE8.5^* mutants, the chambers corresponding to the outflow tract, primitive ventricle, and primitive atrium are clearly delineated (Figure 9b; Figure 10c,d). In contrast, in *Aldh1a2^−/−^* KO mutants the heart chambers are ill-defined due to a failure of cardiac looping [33,37,40]. This difference can most probably be explained by an action of ATRA-activated RAR on cardiogenic mesoderm as early as E7.5, i.e., well-prior to the appearance of the heart tube [41] (Figure 12b).

The aortic sac is a blood collector whose morphology changes during embryonic development together with the pattern of the aortic arches that it supplies [22]. Failure of its partitioning results in a PTA, which is a constant feature of *Rara*^−/−^;*Rarb*^−/−^ and *Rara*^−/−^;*Rarg*^−/−^ KO mutants, and is always associated with various abnormalities of the great arteries derived from the aortic arches [11,12]. A PTA is also constant in *Rarabg^ΔE9.5^* mutants, while it is only rarely observed in *Rarabg^ΔE10.5^* mutants and never found in *Rarabg^ΔE11.5^* mutants ([3,4] and the present report). Therefore, the critical time period during which the RAR signalling is required for septation of the aortic sac is between E9.5 and E10.5 (Figure 12b), which is at least 24 h prior to the appearance of the aorticopulmonary septum at E11.5. This critical time period also precedes the formation of the 6th aortic arches at E10.5, which is logical since the development of the 6th aortic arches is normally required for the appearance of the aorticopulmonary septum [22]. The aorticopulmonary septum and 6th aortic arches all derive from the most caudal portion of the aortic sac. Therefore, the early determination of the PTA and its constant association with the agenesis of the 6th arches (or of their derivative, the arterial duct) in RAR loss-of-function mutants could reflect the absence of a RAR-dependent cell population that normally contributes to the caudal portion of the aortic sac.

The phenotype of *Rarabg^ΔE9.5^* mutants also indicates that RAR signalling is required for the establishment of a bilaterally symmetrical arterial pattern from the 3rd, 4th, and 6th aortic arches. In contrast, this RAR signalling is no longer involved in the subsequent remodeling of this bilateral pattern to produce the definitive arteries because *Rarabg^ΔE10.5^* mutants never display defects of the arch of the aorta, of the right subclavian artery, and of the common carotids.

We have recently shown that RAR signalling between E10.5 and E11.5 is essential for vascular remodeling processes occurring in the caudal portion of the aorta [4]. This process appeared equally deficient in *Rarabg^ΔE9.5^* mutants (present report), yielding a single umbilical artery and absence of the primitive iliac arteries (Figure 12b).

### 4.3. Distinct Timing of RAR Signalling Pathway in Pharyngeal Pouches

A majority of *Rara*^−/−^;*Rarb*^−/−^ KO mutants exhibit a reduced development of the 3rd pharyngeal pouch, which is manifested at later developmental stages by thymic hypoplasia and agenesis. The reduced development of the 3rd pouch is neither more frequent nor more severe in *Rarabg^ΔE9.5^* mutants. This can be accounted for by the timing of RAR excision which is after the onset of budding of the 3rd pouch from the pharyngeal endoderm. Along this line, pharmacologically blocking RAR signalling in cultured mouse embryos impairs the formation of the 3rd pouch at the equivalent of E8.5–E9.5, but not thereafter [42].

The 4th pharyngeal pouch never forms in *Rarabg^ΔE9.5^* mutants, as opposed to *Rara*^−/−^;*Rarb*^−/−^ KO mutants which display a reduced development of this pouch only occasionally [11,12,43]. This difference can be accounted for by a partial functional compensation by RARG in the KO mutants. Altogether, these data indicate that formation of the 3rd and 4th pouches are equally critically dependent on RAR signalling, although at different time points (Figure 12c).

### 4.4. Different Events in Eye Morphogenesis Require a Functional RAR Signalling Pathway

It has been established, for more than 20 years, that impaired RAR functioning in *Rara*^−/−^;*Rarg*^−/−^ and *Rarb*^−/−^;*Rarg*^−/−^ KO mutants results, during late foetal stages (i.e., E14.5 and E18.5), in a large variety of ocular abnormalities affecting the lens, ventral retina, optic fissure, cornea, conjunctiva, and eyelids [12,13]. However, due to the remaining compensatory activity of a RAR isotype still present in these KO mutants, it could not be determined at which developmental stages the RAR pathway is required for the development of the different eye structures.

Our analysis of *Rarabg^iΔE8.^*^5^ mutants indicates that lens agenesis, which was previously reported in a minority of *Rara*^−/−^;*Rarg*^−/−^ KO mutants, is determined between E8.5-E9.5 (Figure 12d), i.e., at the developmental stage when the lens placode is specified from the ectoderm [44]. Contrasting with the low frequency of lens agenesis, severe forms of retinal coloboma affect all *Rara*^−/−^;*Rarg*^−/−^ KO mutants, while a milder form of coloboma, restricted to the optic disc, is present in a majority of *Rarb*^−/−^;*Rarg*^−/−^ KO mutants. The conditional *Rar* knockout approach that we used in the present study indicates that RAR signalling is required for the closure of the optic fissure between E9.5 and E10.5 at the optic cup, then between E10.5 and E11.5 at the optic disc (Figure 12d).

As for the shortening of the ventral retina and the cryptophthalmos, they are as severe in *Rarabg^ΔE9.^*^5^ as in *Rarabg^ΔE10.^*^5^ mutants, while mild in *Rarabg^ΔE11.5^* mutants, in which excision of the 3 *Rar* is obtained at E11.5 ([3,4] and the present report). Therefore, E10.5 to E11.5 spans a critical developmental period during which the RAR signalling pathway is required for the growth of the ventral retina and for the normal distribution of the mesenchyme around the optic cup and lens (Figure 12d).

Eye morphogenesis results from complex cellular interactions between the neurectoderm (which give rise to retina and optic nerve), the surface ectoderm (which give rise to the lens and to the epithelia of the cornea, conjunctiva, and eyelids), and the neural crest cells-derived periocular mesenchyme (which forms the choroid, sclera, and corneal stroma) [45]. As demonstrated previously, ablation of the 3 RARs in the neural crest cells only (yielding *Rara/b/g^NCC−/−^* mutants) recapitulates the eye malformations described in *Rara*^−/−^;*Rarg*^−/−^ KO and *Rarb*^−/−^;*Rarg*^−/−^ KO mutants except for lens agenesis [46]. The fact that the spectrum of ocular defects found in *Rarabg^ΔE9.5^* is not more pronounced than that found in *Rara/b/g^NCC−/−^* mutants implies that the neural crest cells are indeed the sole, direct targets of RAR signalling during eye morphogenesis from E9.5 onward.

### 4.5. A Crucial Role of the RAR Signalling Pathway before the Otocyst Stage

Both excess ATRA and inhibition of RAR signalling have been reported to dramatically alter inner ear morphogenesis in vertebrates [47,48,49,50]. For instance, in *Rara*^−/−^;*Rarg*^−/−^ KO mutants analysed at birth, the inner ear is found as a small epithelial sac without vestibular and auditory divisions [50]. Moreover, both RAR and ATRA-synthesising or degrading enzymes display very dynamic spatial and temporal changes in their expression patterns during inner ear development [50,51]. These previous observations suggested to us that signalling through RAR could play a major role in the transformation of a simple vesicle, the otocyst, into a sophisticated system, the membranous labyrinth, which is essential for the senses of equilibrium and hearing. The reduced size of both the endolymphatic sac and saccule we found in *Rarabg^Δ9.5^* and *Rarabg^Δ10.5^* mutants is similar to that previously reported in *Aldh1a3^−/−^* KO mutants [14]. This indicates that ATRA-activated RARs are involved from E9.5 in no more than fine-tuning the growth of the membranous labyrinth. Accordingly, the dramatic failure of inner ear morphogenesis that we previously described in *Rara*^−/−^;*Rarg*^−/−^ KO mutants is necessarily determined at (or before) the otocyst stage (Figure 12d). This conclusion is further supported by the obvious hypoplasia of the otocyst that we found in *Rarabg^iΔ8.5^* mutants, and is compatible with a role of RAR signalling in conferring regional identities to the inner ear around E8.5 [49,52].

Along these lines, the supernumerary, ectopic otocysts in *Rarabg^iΔ8.5^* mutants are also observed in *Rara*^−/−^;*Rarb*^−/−^ and *Rara*^−/−^;*Rarg*^−/−^ KO mutants. They likely reflect a role of RAR signalling in the specification of the otogenic field [43,53].

### 4.6. Early Functions of the RAR Signalling Pathway in the Midface

A midline facial cleft from E11.5 is a constant feature of *Rara*^−/−^;*Rarg*^−/−^ KO mutants and *Rara/b/g^NCC−/−^* mutants [13,54]. Our study shows that *Rarabg^Δ9.5^* mutants exhibit alterations in facial morphology related to this abnormal clefting process as early as E10.5. They further demonstrate that the critical period during which the RARs are instrumental to midfacial morphogenesis is between E9.5 and E10.5 (Figure 12f), i.e., shortly after the completion of neural crest cell emigration from the forebrain (E9.0 [55]), but well before the merging of the medial nasal processes (E11.5 to E12.0). The fact that the midfacial clefts found in *Rarabg^ΔE9.5^* mutants are not more severe than those found in *Rara/b/g^NCC−/−^* mutants implies again that the neural crest cells are indeed the sole, direct targets of RAR signalling during facial morphogenesis from E9.5 onward. 

The nasal cavities are embedded in the neural crest cell-derived midfacial mesenchyme. In *Rarb*^−/−^;*Rarg*^−/−^ KO mutants and in *Aldh1a3^−/−^* KO mutants, their morphogenesis and communication with the oral cavity are impaired without association with a midfacial cleft [12,15]. These data and the results of our present study indicate that the action of ATRA-activated RARB and/or RARG is instrumental to the development of nasal cavities between E10.5 and E11.5 (Figure 12f).

### 4.7. The RAR Signalling Pathway Is Instrumental to Axial Rotation

Axial rotation, a vast morphogenetic event whose mechanism is still unclear, is impaired in some *Rara*^−/−^;*Rarg*^−/−^ KO mutants [53]. Axial rotation is also impaired in *Rarabg^ΔE8.5^* mutants, with a complete penetrance, further demonstrating that RAR signalling is instrumental to this process (Figure 12g). *Rarabg^ΔE8.5^* mutants also display the *other* abnormalities found in *Rara*^−/−^;*Rarg*^−/−^ KO mutants, including shortening of the anterior-posterior axis, small and densely packed somites, and persistent opening of the rhombencephalic neural tube [53]. 

Interestingly, the other abnormalities displayed by *Rarabg^ΔE8.5^* and *Rarabg^iΔE8.5^* mutants have been reported in *Aldh1a2^−/−^* KO mutants. This is the case of absence of the sinus venosus, absence of large vitelline vessels, agenesis of the dorsal pancreatic bud, and bilateral lung agenesis [33,34,37,56,57,58]. Unfortunately, it is not possible to study the outcome of these anomalies at developmental stages beyond E10.5 because ablation of all 3 RARs at E8.5 results in early growth arrest and embryonic lethality. 

### 4.8. RAR Signalling in the Development of the Urogenital System and Rectum

We have previously shown that *Rarabg^ΔE10.5^* mutants at E12.5 to E14.5 constantly display an agenesis of the rectum together with defects in the urogenital system, including: complete agenesis of the Müllerian ducts, abnormal endings of the Wolffian ducts, small and ectopic kidneys, and hypoplasia of the cloaca [4]. The formation of the rectum and Müllerian ducts could not be addressed in *Rarabg^ΔE9.5^* mutants due to their early death, but those analysed at E12.5 displayed the other malformations with the same severity. This observation strengthens our previous conclusion that E10.5 to E11.5 spans a critical window of time for RAR signalling in the development of the rectum and urogenital system [4] (Figure 12h). 

Uni- or bilateral absence of any kidney tissue (i.e., kidney agenesis) is observed in a majority of *Rara*^−/−^;*Rarg*^−/−^ KO mutants analysed at birth. In contrast, the first anlage of the kidney (metanephric mesenchyme), whose development initiates at E10.5, is present bilaterally in all *Rara*^−/−^;*Rarg*^−/−^ KO mutants at E11.5 and E12.5 [11,59]. In this context, it is interesting to note that this mesenchyme is also identified in E11.5 and E12.5 *Rarabg^ΔE9.5^* and *Rarabg^ΔE10.5^* mutants [4]. This finding indicates that the RAR signalling pathway is definitely not required in specifying the metanephric mesenchyme, but instead could be involved in its survival [11]. 

### 4.9. Study Limitations

In the present study, we established a list of critical time-windows requiring a functional RAR signalling pathway in the development of the cardiovascular and respiratory systems, sense organs, and facial structures (Figure 12a–h). This list provides a solid basis for further studies aimed at elucidating the cellular and molecular mechanisms controlled by RARs in the embryo. It is, however, not exhaustive. In fact, the early lethality of *Rarabg^ΔE8.5^* and *Rarabg^ΔE9.5^* mutants makes it impossible to trace the origin of several congenital anomalies observed in compound *Rar* KO mutants at birth [1,2]. Moreover, insofar as we have studied the consequences of the inactivation of the *Rar* genes in embryos from E9.5 to E14.5, our data do not allow us to rule out important functions of RARs in organ growth and histogenesis during the late foetal and perinatal periods.

## Figures and Tables

**Figure 1 biomedicines-11-00198-f001:**
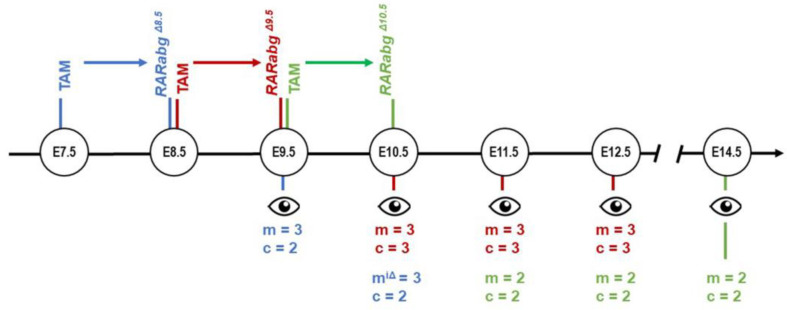
Correspondence between the day of TAM treatment, the time lapse for obtaining mutants with loss of RARA and RARG (colored arrows), the schedule of the phenotypic analyses, and the number of mutants (m) and control (c) embryos examined by HREM at each developmental stage. m^iΔ^, these mutants were incompletely excised for *Rara* and *Rarg*; see the main text for further details.

**Figure 2 biomedicines-11-00198-f002:**
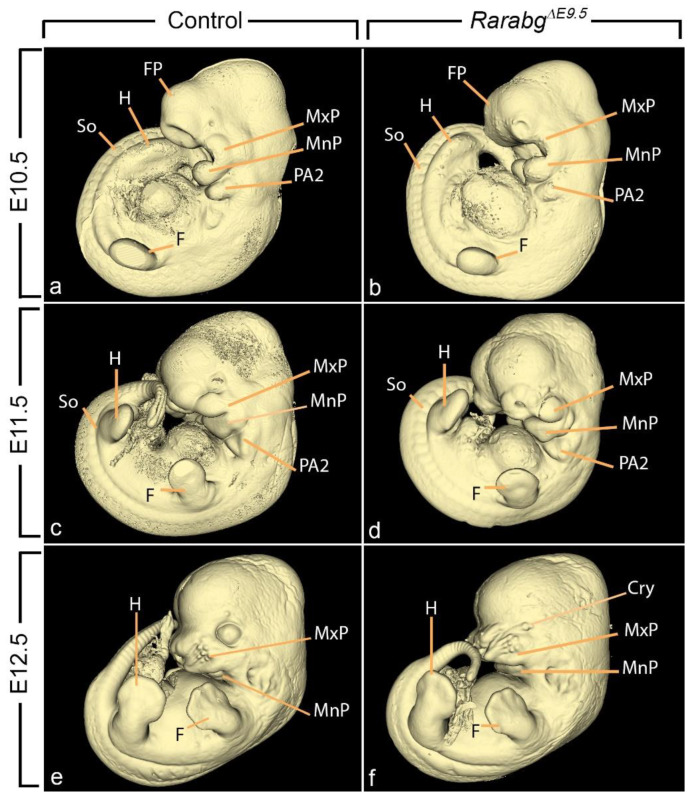
Left lateral views of control embryos (**a**,**c**,**e**) and *Rarabg^ΔE9.5^* mutant littermates (**b**,**d**,**f**) at E10.5, E11.5, and E12.5, as indicated. Cry, cryptophthalmos; F, forelimb bud; FP, frontonasal process; H, hindlimb bud; MxP and MnP, maxillary and mandibular processes of the 1st pharyngeal arch, respectively; PA2, 2nd pharyngeal arch. So, somite. Same magnification in (**a**–**f**).

**Figure 3 biomedicines-11-00198-f003:**
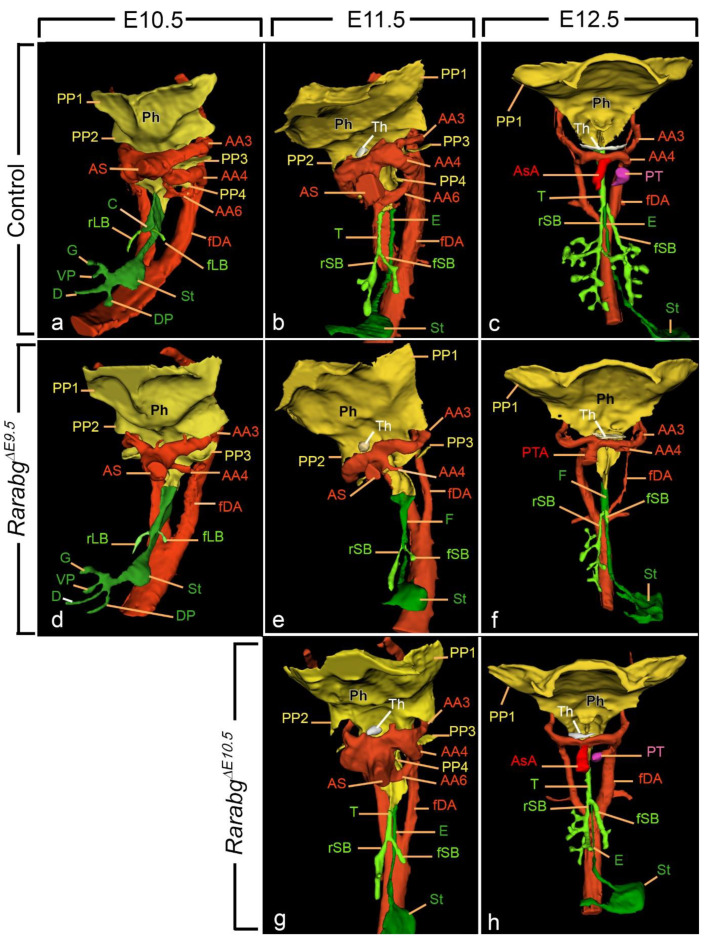
Left latero-ventral views (**a**,**b**,**d**,**e**,**g**) and ventral views (**c**,**f**,**h**) of 3D-reconstructions of the pharyngeal and foregut regions in control embryos (**a**–**c**), *Rarabg^ΔE9.5^* mutants (**d**–**f**) and *Rarabg^Δ10.5^* mutants (**g**,**h**) at E10.5, E11.5, and E12.5, as indicated. AA3, AA4, and AA6, 3rd, 4th, and 6th aortic arches; AS, aortic sac; AsA, ascending aorta, C, presumptive carina; D, duodenum; fDA, left dorsal aorta; DP, dorsal pancreatic bud; E, esophagus F, (undivided) foregut tube; G, gallbladder; rLB and fLB, lumens of the right and left lung buds, respectively; Ph, pharyngeal cavity; PP1, PP2, PP3, and PP4, 1st, 2nd, 3rd, and 4th pharyngeal pouches; PT, pulmonary trunk; PTA, persistent truncus arteriosus; rSB and fSB, right and left stem bronchi, respectively; St, stomach; T, trachea; Th, thyroid gland; VP, ventral pancreatic bud. Same magnifications in (**a**,**d**), in (**b**,**e**,**g**) and in (**c**,**f**,**h**).

**Figure 4 biomedicines-11-00198-f004:**
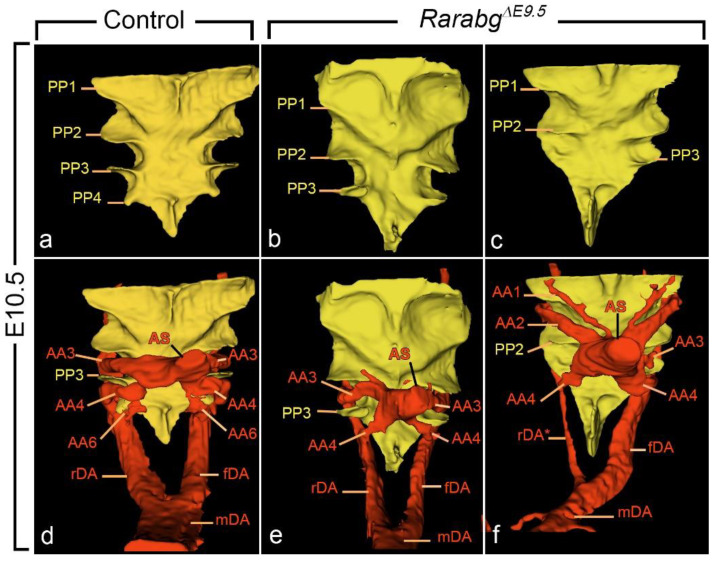
Ventral views of 3D-reconstructions of the pharyngeal cavity (**a**–**f**) and aortic arches (**d**–**f**) in a control embryo (**a**,**d**) and in *Rarabg^ΔE9.5^* mutants (**b**,**c**,**e**,**f**) at E10.5. AA1, AA2, AA3, AA4, and AA6, 1st, 2nd, 3rd, 4th, and 6th aortic arches; AS, aortic sac; rDA, fDA and mDA, right, left and midline dorsal aortas, respectively; rDA*, hypoplastic right dorsal aorta; PP1, PP2, PP3, and PP4, 1st, 2nd, 3rd, and 4th pharyngeal pouches. Same magnifications in (**a**–**f**).

**Figure 5 biomedicines-11-00198-f005:**
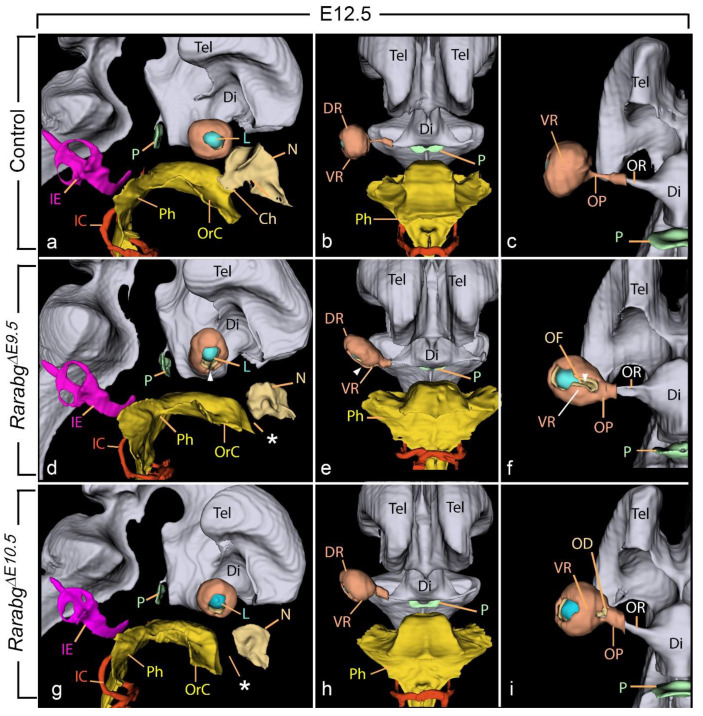
Right lateral views (**a**,**d**,**g**), frontal views (**b**,**e**,**h**) and ventral views (**c**,**f**,**i**) of 3D-reconstructions of the head region in a control embryo (**a**–**c**), in a *Rarabg^ΔE9.5^* mutant (**d**–**f**), and in a *Rarabg^Δ10.5^* mutant (**g**–**i**) at E12.5. The panel focuses: (i) on the anatomy of the optic cup and on its position relative to its neighbouring structures and (ii) on the communications between the nasal and oral cavities (the primitive choanae) in control embryos which is absent in mutant embryos, leading to choanal atresia. Ch, primitive choana; Di, cavity of the diencephalon (3rd ventricle); DR and VR, dorsal and ventral region of the retina, respectively; IC, internal carotid artery; IE, cavities of the inner ear; L, lens; N, nasal cavity; OrC, oral cavity; OD, optic disc; OF, optic fissure; OP, optic peduncle; OR, optic recess of the 3rd ventricle; P, pituitary gland; Ph, pharyngeal cavity; Tel, telencephalic vesicle. The white arrowhead points to the ventral opening in the optic cup (retinal coloboma) and the asterisks indicate the “gap” between the nasal and oral cavities resulting from the absence of the choana. Same magnifications in (**a**,**d**,**g**), in (**b**,**e**,**h**) and in (**c**,**f**,**i**).

**Figure 6 biomedicines-11-00198-f006:**
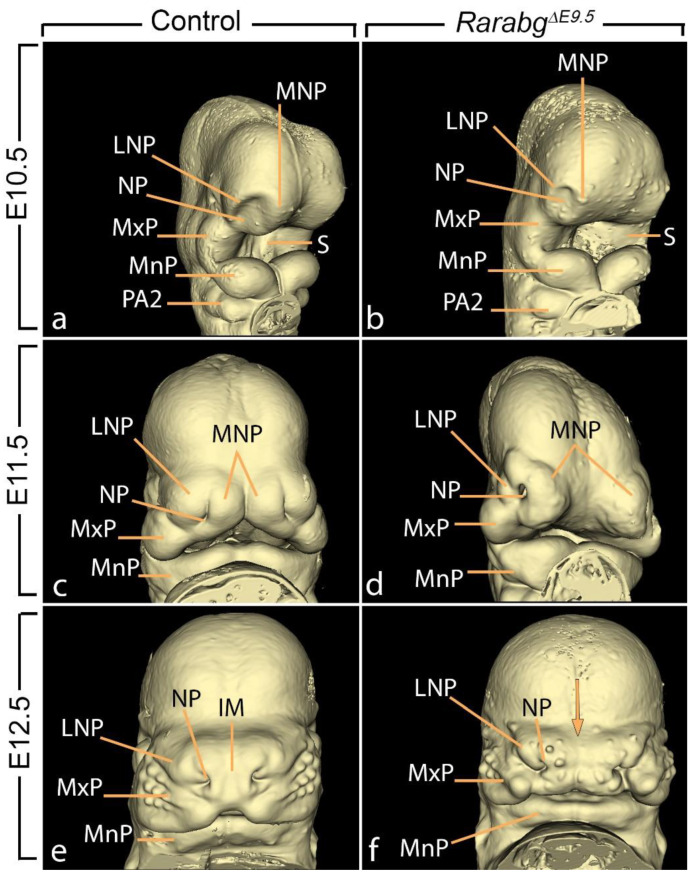
Facial aspect of control embryos (**a**,**c**,**e**) and *Rarabg^ΔE9.5^* mutant littermates (**b**,**d**,**f**) at E10.5, E11.5, and E12.5, as indicated. IM, intermaxillary segment; LNP and MNP, lateral and medial nasal processes, respectively; MxP and MnP, maxillary and mandibular processes of the 1st pharyngeal arch, respectively; NP, nasal pit; PA2, second pharyngeal arch; S, stomodeum (primitive mouth). The arrow (in f) points to the midfacial cleft. Same magnifications in (**a**,**b**), in (**c**,**d**) and in (**e**,**f**).

**Figure 7 biomedicines-11-00198-f007:**
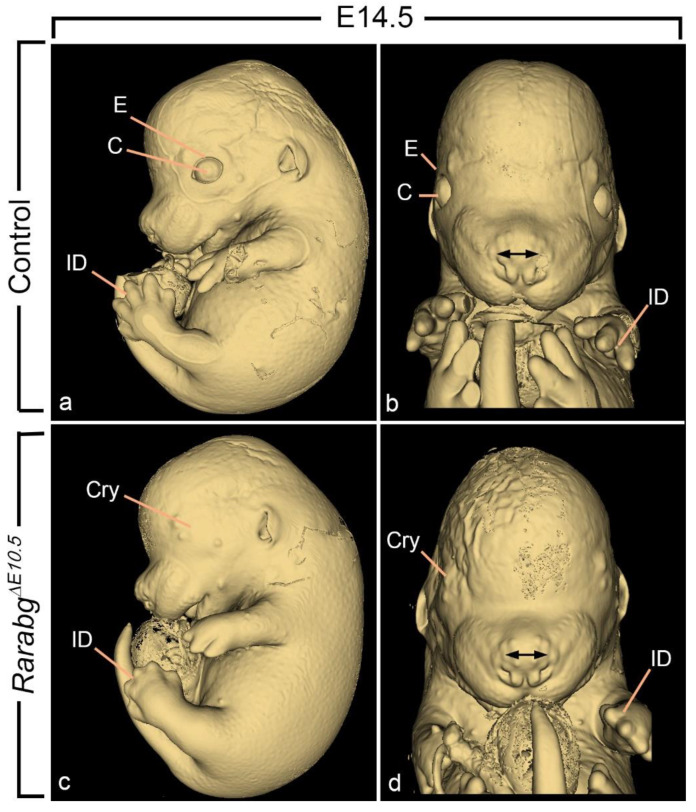
External appearance of control (**a**,**b**) and *Rarabg^ΔE10.5^* mutant (**b**,**d**) littermates at E14.5. The cryptophtalmos (Cry), the failure of separation of the digits, and the shortening of the snout are characteristic of the *Rarabg^ΔE10.5^* mutant phenotype (Mark et al., 2021). C, cornea; E, upper eyelid fold; ID, interdigital space. The double arrow indicates the spacing of the nostrils. Same magnifications in (**a**,**c**) and in (**b**,**d**).

**Figure 8 biomedicines-11-00198-f008:**
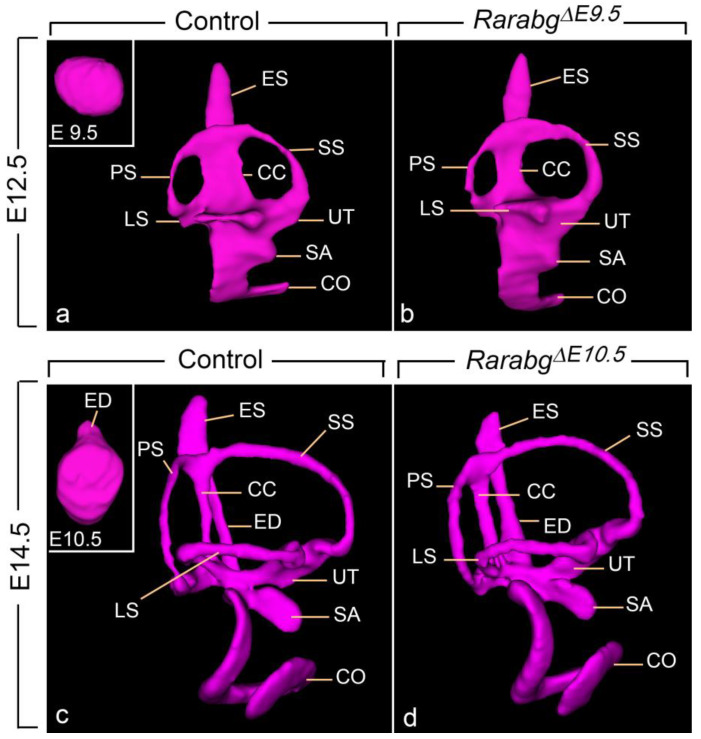
Lateral views of 3D-reconstructions of the right membranous labyrinth in control embryos (**a,c**), in a *Rarabg^ΔE9.5^* mutant (**b**) and in a *Rarabg^Δ10.5^* mutant (**d**) at E12.5 and E14.5, as indicated. The insets in (**a**,**c**) represent lateral views of normal otocysts at E9.5 and E10.5, respectively. CC, common crus; CO, cochlea; ED, endolymphatic duct; ES, endolymphatic sac; LS, lateral semicircular canal; PS, posterior semicircular canal; SA, saccule; SS, superior semicircular canal; UT, utricle. Same magnification in (**a**–**d**).

**Figure 9 biomedicines-11-00198-f009:**
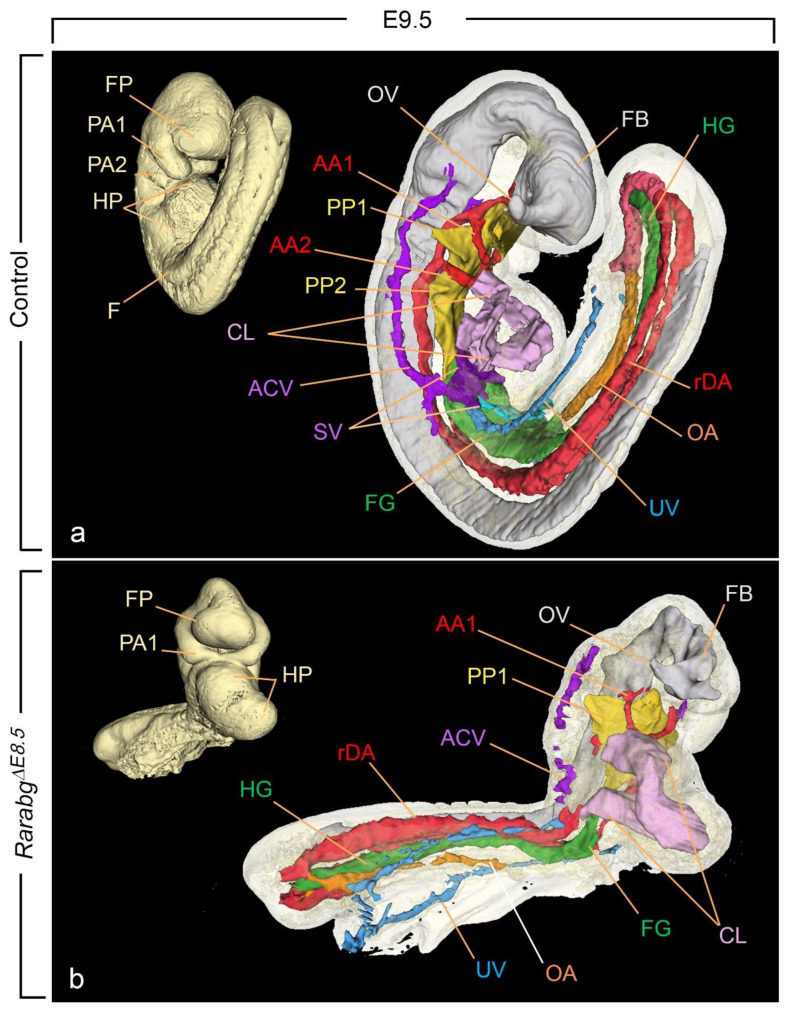
Right latero-ventral views of the external and internal aspects of a control embryo (**a**) and of a *Rarabg^ΔE8.5^* mutant littermate (**b**) at E9.5. AA1 and AA2, 1st and 2nd aortic arches; ACV, anterior cardinal vein; CL, cardiac loop; rDA, right dorsal aorta; F, forelimb bud; FB, forebrain; FG, foregut; FP, frontonasal process; HG, hindgut; HP, heart prominence; OA, omphalomesenteric (vitelline) artery; OV, optic vesicle; PA1 and PA2, 1st and 2nd pharyngeal arches; PP1 and PP2, 1st and 2nd pharyngeal pouches; UV, umbilical vein; SV, sinus venosus. Same magnification in (**a**,**b**).

**Figure 10 biomedicines-11-00198-f010:**
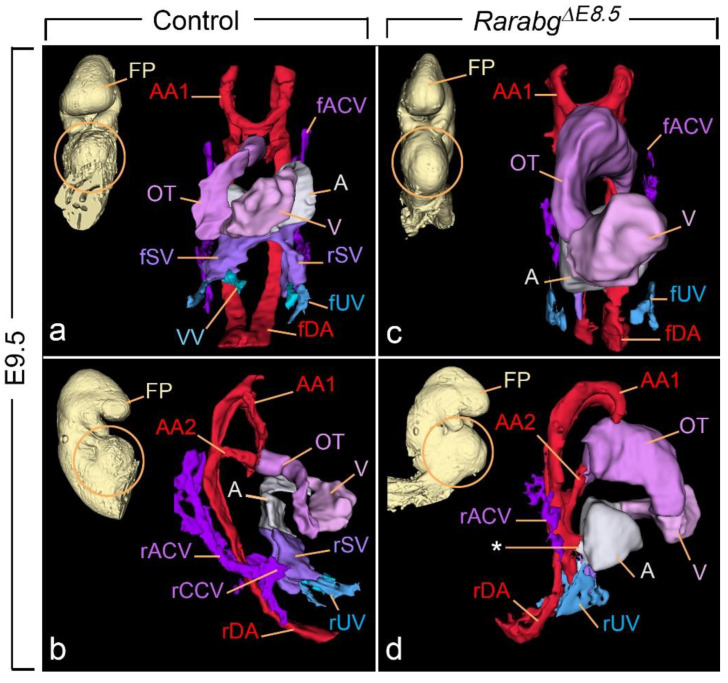
Frontal views (**a**,**c**) and right lateral views (**b**,**d**) of 3D-reconstructions of the heart region (encircled on the insets) of a control embryo (**a**,**b**) and of a *Rarabg^ΔE8.5^* mutant littermate (**c**,**d**) at E9.5. A, primitive atrium; AA1 and AA2, 1st and 2nd aortic arches, respectively; fACV and rACV, left and right anterior cardinal veins, respectively; rCCV, right common cardinal vein; fDA and rDA, left and right dorsal aortas, respectively; FP, frontonasal process; fSV and rSV, left and right horns of the sinus venosus; fUV and rUV, left and right umbilical veins, respectively; V, primitive ventricle. The asterisk indicates an aorto-atrial fistula. Same magnification in (**a**–**d**).

**Figure 11 biomedicines-11-00198-f011:**
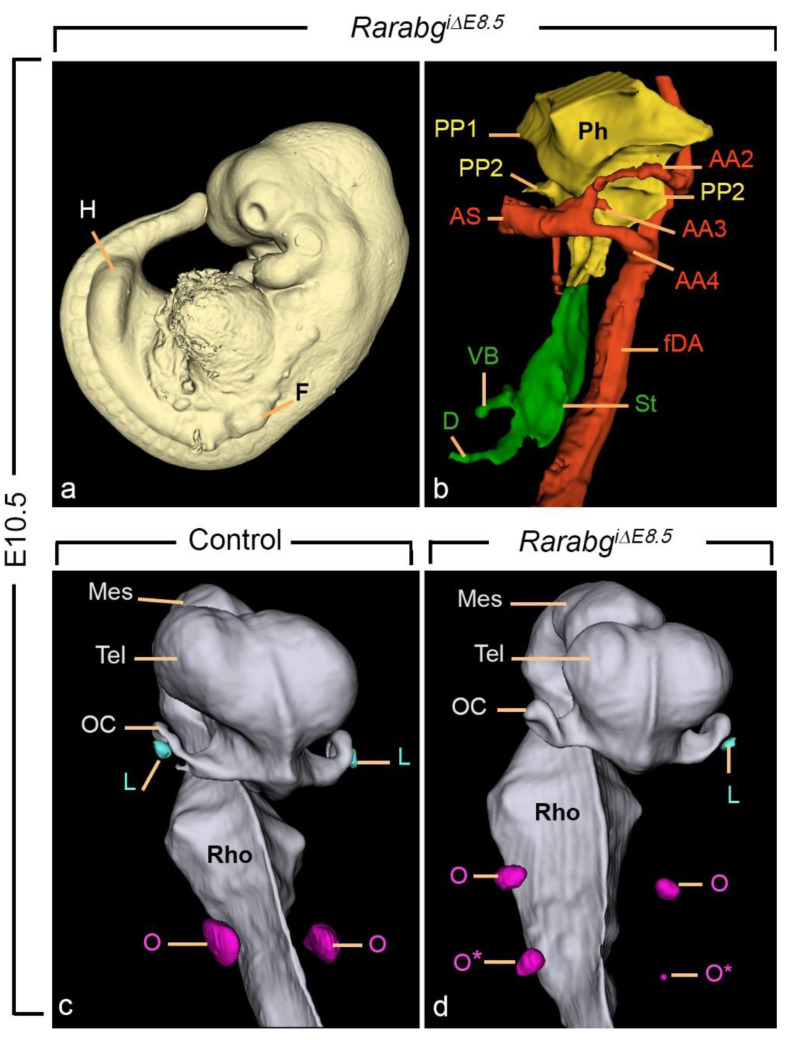
Anatomical features characteristic of the *Rarabg^iΔE8.5^* mutant phenotype at E10.5. (**a**) Left lateral view of the external aspect. Note the small size of the forelimb bud (F), as opposed to the normal appearance of the hindlimb bud (H) and compare with Figure 2a. (**b**) Left lateral view of a 3D-reconstruction of the pharyngeal and foregut regions. Compare with Figure 3a and note: (i) the abnormal persistence of the 2nd aortic arch (AA2) and (ii) the absence of the dorsal portion of the 3rd aortic arch (AA3), of the 6th aortic arch, of the 3rd and 4th pharyngeal pouches, and of the dorsal pancreatic bud. (**c**,**d**) Ventral views of 3D-reconstructions of the brain region of a control embryo (**c**) and of a *Rarabg^iΔE8.5^* mutant (**d**) at E10.5. Note the marked hypoplasia of the mutant otocysts (O), the presence of ectopic otocysts (O*), and the absence of the lens (L) on the right side. AA2, AA3, and AA4, 2nd, 3rd, and 4th aortic arches; AS, aortic sac; fDA, left dorsal aorta; D, duodenum; Mes, cavity of the mesencephalon; OC, optic cup; Ph, pharyngeal cavity; PP1 and PP2, 1st and 2nd pharyngeal pouches; Rho, cavity of the rhombencephalon; St, stomach; VB, ventral bud of the foregut; Tel, telencephalic vesicle. Same magnifications in (**c**,**d**).

**Figure 12 biomedicines-11-00198-f012:**
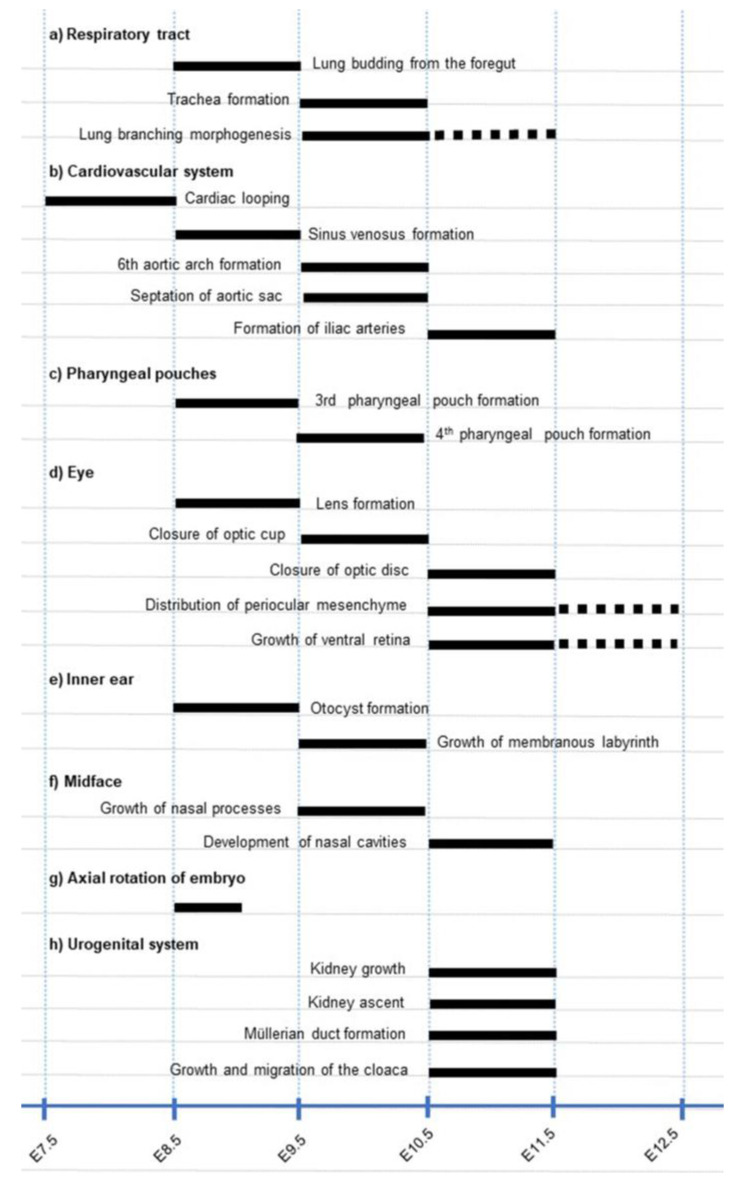
Critical periods in development requiring a functional RAR signalling pathway. The developmental processes analysed in *Rarabg^ΔE8.5^, Rarabg^ΔE9.5^, Rarabg^ΔE10.5^* and *Rarabg^ΔE11.5^* mutants are listed. Solid bars indicate the periods when impaired signalling through RARs causes major structural defects and stippled bars indicate the periods when the defects appear less frequent or less severe. For further details, see the main text and [4].

**Table 1 biomedicines-11-00198-t001:** Abnormalities displayed by *Rarabg^ΔE9.5^* mutants analysed by HREM at E10.5, E11.5, and/or E12.5 (*n* = 3 at each developmental stage). # These abnormalities are completely penetrant. § These abnormalities are equal in frequency and severity to those present in *Rarabg^ΔE10.5^* mutants [4]. * From the phenotypes of *Rara*^−/−^;*Rarb*^−/−^, *Rara*^−/−^;*Rarg*^−/−^ and *Rarb*^−/−^;*Rarg*^−/−^ KO mutants at late foetal stages [11,12,13]. ** From the phenotypes of *Rarabg^ΔE10.5^* mutants [4]. *** From the phenotype of *Aldh1a3^−/−^* KO mutants at late foetal stages [14,15].

	Predicted Outcome
Respiratory defects	
No left lung bud	Left lung agenesis *
Delayed formation of secondary bronchi #	Lung hypoplasia *
Agenesis of the trachea #	
**Cardiovascular defects**	
Persistent truncus arteriosus #	
Abnormal patterning of the aortic arches #	Abnormal cephalic arteries *
No remodelling of umbilical roots # §	Absent iliac arteries **
**Ocular defects**	
Cryptophthalmos # §	No cornea, iris and eyelid folds *
Short ventral retina # §	
Ventral rotation of the optic cup and lens # §	
Absence of closure of the optic fissure #	Colobomatous microphtalmia*
**Inner ear defects**	
Reduced development of the saccule and endolymphatic sac #	Hypoplasia of the saccule and endolymphatic sac ***
**Facial defects**	
Hypoplastic nasal processes #	Midfacial cleft *
Persistent oronasal membrane # §	Choanal atresia ***
Small nasal cavities # §	
**Urogenital defects**	
• Hypoplastic urogenital sinus # §	Agenesis of the rectum **
• Kidney hypoplasia # §	
• Kidney ectopia # §	
• Abnormal endings of Wolffian ducts # §	

**Table 2 biomedicines-11-00198-t002:** Development of the aortic arches and pharyngeal pouches on the left and right sides (L and R) of individual *Rarabg^ΔE9.5^* mutants at E10.5 (a–c) and at E11.5 (d–f), and of control embryos. AA1 to AA6, aortic arches 1 to 6 (+: complete, the aortic arch connects the aortic sac to the dorsal aorta; −: absent; +/−: missing dorsal part, the aortic arch is not connected to the aorta). PP3 and PP4, 3rd and 4th pharyngeal pouches (pre, present and of normal size; hy, hypoplastic; abs, absent).

	Individual Mutant Embryos												Control Embryos			
	E10.5						E11.5						E10.5		E11.5	
	a		b		c		d		e		f		(n = 3)		(n = 3)	
	L	R	L	R	L	R	L	R	L	R	L	R	L	R	L	R
AA1	+/−	+/−	−	−	−	−	−	−	−	−	−	−	−	−	−	−
AA2	+	+	+	+	−	−	−	−	−	−	−	−	−	−	−	−
AA3	+/−	−	+/−	−	+	+	+	+	+	+	+/−	−	+	+	+	+
AA4	+	+/−	+	+	+	+	+	+	+/−	+/−	−	+	+	+	+	+
AA6	−	−	−	−	−	−	−	−	−	−	−	−	+	+	+	+
PP3	hy	abs	hy	abs	pre	pre	pre	pre	pre	pre	hy	hy	pre	pre	pre	pre
PP4	abs	abs	abs	abs	abs	abs	abs	abs	abs	abs	abs	abs	pre	pre	pre	pre

**Table 3 biomedicines-11-00198-t003:** Comparison of the abnormalities displayed by *Rarabg^ΔE8.5^* mutants (n = 3) and *Aldh1a2^−/−^* KO mutants analysed at E9.5. +, Present; −, Absent. # These abnormalities are completely penetrant. NA, not applicable as the invagination of the otic placode was still underway in all these mutants; NR not reported. * From [33,34].

Type of Defect	*Rarabg^ΔE8.5^*	*Aldh1a2^−/−^* KO *
Failure of axial rotation	+ #	+
Absence of the limb buds	+ #	+
Open ventral body wall	+ #	+
Small and densely packed somites	+ #	+
Narrow and folded neural tube	+ #	+
Open neural tube	+	+
Absence of the 2nd aortic arch	+	+
Dilated heart	+ #	+
Absence of the sinus venosus	+ #	+
Absence of large vitelline veins	+	+
Hypoplastic otocyst	NA	+
No looping of the heart tube	−	+
Aorto-atrial fistula	+ #	NR

## Data Availability

All data needed to evaluate the conclusions in the paper are present in the paper and/or the Appendix A.

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
