# Peer review of "Timeline of Developmental Defects Generated upon Genetic Inhibition of the Retinoic Acid Receptor Signaling Pathway"

_biomedicines, 2023, doi:10.3390/biomedicines11010198_

Round 1

Reviewer 1 Report

In this study, Marius Teletin and colleagues examined the role of RARs (retinoic acid receptors) in the early development of mice using HREM technique. They selectively deleted RARs at different stages of development and found that RARs are essential for axial rotation and the formation of organ and limb progenitors from E8.5 to E9.5, and for lung branching, tracheal development, and the formation of the inner ear, facial structures, and arteries from E9.5 onwards. These findings highlight the vital role of RARs in a range of tissue morphogenesis processes during early mouse development. The research is of high quality and the authors have done an excellent job in their work. However, it is noteworthy to mention that RA signaling is required for the specification and differentiation of spinal cord progenitors from bi-potential Neuromesodermal progenitors which are believed to be specified from E7.5 stages. Deleting RARs slightly earlier than E7.5 (Tam from the E6.5 stages) and monitoring the phenotypes at later stages would address their roles in early gastrulation stages as well. 

Reviewer 2 Report

The manuscript from Teletin et al. provides a very extensive and convincing characterization of the essential actions of the three retinoic acid receptors (RARα, -β and -γ) in mouse embryogenesis, during the period ranging from E7.5 through E11.5.  Data are reported which establish critical time-windows that require functional RAR-signaling for the cardiovascular and respiratory systems, sense organs and facial structures.  The work appears to be of high technical quality and all of the data are believable.  The manuscript is very well-written and was a pleasure to read.  The conclusions the authors draw was derived directly from their data.  Overall, this is a very strong contribution from a leading lab exploring RAR-signaling during embryogenesis.